# Levels of pneumococcal conjugate vaccine coverage and indirect protection against invasive pneumococcal disease and pneumonia hospitalisations in Australia: An observational study

Jocelyn Chan[1,2]*, Heather F. Gidding[3,4,5], Christopher C. Blyth[6], Parveen Fathima[7], Sanjay Jayasinghe[5,8], Peter B. McIntyre[5], Hannah C. Moore[7], Kim Mulholland[1,2,9], Cattram D. Nguyen[1,2], Ross Andrews[10,11], Fiona M. Russell[1,2]

1 Asia-Pacific Health Research Group, Murdoch Children's Research Institute, Melbourne, Australia, 2 Department of Paediatrics, The University of Melbourne, Melbourne, Australia, 3 Northern Clinical School, The University of Sydney, Sydney Australia, 4 Women and Babies Health Research, Kolling Institute, Northern Sydney Local Health District, Sydney Australia, 5 National Centre for Immunisation Research and Surveillance of Vaccine Preventable Diseases, The Children's Hospital at Westmead, Sydney, Australia, 6 School of Medicine, University of Western Australia, Perth, Australia, 7 Wesfarmers Centre of Vaccines and Infectious Diseases, Telethon Kids Institute, University of Western Australia, Perth, Australia, 8 Children's Hospital at Westmead Clinical School, Faculty of Medicine, University of Sydney, Sydney, Australia, 9 Department of Infectious Disease Epidemiology, London School of Hygiene and Tropical Medicine, London, United Kingdom, 10 Global & Tropical Health Division, Menzies School of Health Research, Charles Darwin University, Darwin, Australia, 11 National Centre for Epidemiology & Population Health, Australian National University, Canberra, Australian Capital Territory, Australia

* jocelyn.chan@mcri.edu.au

## Abstract

### Background

There is limited empiric evidence on the coverage of pneumococcal conjugate vaccines (PCVs) required to generate substantial indirect protection. We investigate the association between population PCV coverage and indirect protection against invasive pneumococcal disease (IPD) and pneumonia hospitalisations among undervaccinated Australian children.

### Methods and findings

Birth and vaccination records, IPD notifications, and hospitalisations were individually linked for children aged <5 years, born between 2001 and 2012 in 2 Australian states (New South Wales and Western Australia; 1.37 million children). Using Poisson regression models, we examined the association between PCV coverage, in small geographical units, and the incidence of (1) 7-valent PCV (PCV7)-type IPD; (2) all-cause pneumonia; and (3) pneumococcal and lobar pneumonia hospitalisation in undervaccinated children. Undervaccinated children received <2 doses of PCV at <12 months of age and no doses at ≥12 months of age. Potential confounding variables were selected for adjustment a priori with the assistance of a directed acyclic graph.

**Data Availability Statement:** The custodians of the data used for this analysis are the New South Wales Centre for Health Record Linkage (https://www.cherel.org.au/) and Western Australia - Data Linkage (https://www.datalinkage-wa.org.au/). Interested researchers may apply at these sites for data access.

**Funding:** This study was funded by the Population Health Research Network Proof of Concept Project, a capability of the Commonwealth Government Collaborative Research Infrastructure Strategy and Education Investment Fund Super Science Initiative, and the Australian National Health and Medical Research Council (project grant GNT1082342, chief investigator HFG). HFG, CCB, HCM and FMR are funded by Australian National Health and Medical Research Council (NHMRC) fellowships. Specifically, FMR is funded by an NHMRC Translating Research into Practice (TRIP) fellowship and Investigator grant. JC is funded by the Australian Research Training Program scholarship. HCM is further supported by a Telethon Kids Institute Emerging Research Leader Fellowship. The funders had no role in study design, data collection and analysis, decision to publish, or preparation of the manuscript.

**Competing interests:** I have read the journal's policy and the authors of this manuscript have the following competing interests: KM is a member of the WHO SAGE Working Group on Pneumococcal Vaccines. CDN is an investigator on a study outside the submitted work that received funding from Pfizer inc. The other authors have declared that no competing interests exist.

**Abbreviations:** aIRR, adjusted incidence rate ratio; ARIA, Accessibility or Remoteness Index of Australia; CI, confidence interval; DAG, directed acyclic graph; IPD, invasive pneumococcal disease; IRSAD, Index of Relative Socio-economic Advantage and Disadvantage; PCV, pneumococcal conjugate vaccine; PCV7, 7-valent PCV; PPV23, 23-valent pneumococcal polysaccharide vaccine; SLA, statistical local area; VT, vaccine-type.

There were strong inverse associations between PCV coverage and the incidence of PCV7-type IPD (adjusted incidence rate ratio [aIRR] 0.967, 95% confidence interval [CI] 0.958 to 0.975, $p$-value < 0.001), and pneumonia hospitalisations (all-cause pneumonia: aIRR 0.991 95% CI 0.990 to 0.994, $p$-value < 0.001) among undervaccinated children. Sub-group analyses for children <4 months old, urban, rural, and Indigenous populations showed similar trends, although effects were smaller for rural and Indigenous populations. Approximately 50% coverage of PCV7 among children <5 years of age was estimated to prevent up to 72.5% (95% CI 51.6 to 84.4) of PCV7-type IPD among undervaccinated children, while 90% coverage was estimated to prevent 95.2% (95% CI 89.4 to 97.8). The main limitations of this study include the potential for differential loss to follow-up, geographical misclassification of children (based on residential address at birth only), and unmeasured confounders.

## Conclusions

In this study, we observed substantial indirect protection at lower levels of PCV coverage than previously described—challenging assumptions that high levels of PCV coverage (i.e., greater than 90%) are required. Understanding the association between PCV coverage and indirect protection is a priority since the control of vaccine-type pneumococcal disease is a prerequisite for reducing the number of PCV doses (from 3 to 2). Reduced dose schedules have the potential to substantially reduce program costs while maintaining vaccine impact.

## Author summary

### Why was this study done?

- Pneumococcal conjugate vaccines (PCVs) reduce the burden of pneumococcal disease in vaccinated and unvaccinated populations through both direct and indirect (herd) effects.

- The indirect effects of a vaccine comprise a substantial component of overall vaccine impact, contributing to the cost-effectiveness of the vaccine, but little is known about what factors contribute to herd protection, including vaccination coverage.

- In this study, we examined associations between PCV coverage and indirect effects within diverse populations in Australia.

### What did the researchers do and find?

- Using a large dataset of 1.3 million children from 2 states in Australia, we quantified the relationship between PCV coverage within small geographical units and indirect protection against pneumococcal disease. We also performed similar analyses for infants too young to be fully vaccinated, urban, rural, and Indigenous populations.

- There were strong inverse relationships between PCV coverage and the incidence of severe invasive disease due to vaccine types and pneumonia hospitalisations among

undervaccinated children, i.e., higher coverage was associated with greater reductions in disease due to indirect effects. We also found substantial indirect effects at relatively low levels of PCV coverage. We estimated that 50% and 90% coverage of 7-valent PCV (PCV7) among children under 5 years of age prevented almost three-quarters (72.5%, 95% confidence interval [CI] 51.6 to 84.4) and almost all (95.2%, 95% CI 89.4 to 97.8) of PCV7-type severe invasive disease, respectively. For pneumonia, we estimated that 50% and 90% coverage was sufficient to prevent one-third (33.3%, 95% CI 27.3 to 38.8) and about half (51.7%, 95% CI 43.7 to 58.6) of all-cause pneumonia hospitalisations among undervaccinated children.

- These trends were similar for children less than 4 months old, urban, rural, and Indigenous populations, although these effects were smaller for rural and Indigenous populations. There was also a trend towards decreasing incidence of PCV13-type IPD among undervaccinated children as PCV13 coverage increased.

## What do these findings mean?

- Our results challenge existing assumptions that high PCV coverage is required to achieve substantial indirect protection.

- Understanding the determinants of indirect effects are particularly urgent as countries that have controlled vaccine-type pneumococcal disease consider using reducing the number of PCV doses (from 3 to 2). Reduced dose schedules have the potential to significantly lower program costs while maintaining vaccine impact, providing indirect protection is achieved and preserved.

## Introduction

Infections due to *Streptococcus pneumoniae*, the pneumococcus, are a leading cause of morbidity and mortality among children globally [1]. Pneumococcal conjugate vaccines (PCVs) have been successful in reducing pneumococcal disease through the direct protection of vaccinated individuals and indirect protection of both vaccinated and unvaccinated individuals [2]. Following the introduction of PCV to childhood immunisation schedules, many high-income countries, including Australia, have observed the near elimination of vaccine-type (VT) invasive pneumococcal disease (IPD) with impacts extending beyond targeted age groups due to reduced transmission of VT pneumococci [3–5].

The 13-valent PCV (Prevnar 13, Pfizer) and 10-valent PCV (Synflorix, GSK) are costly, and infant immunisation schedules are becoming increasingly crowded due to multiple vaccines. Therefore, a PCV schedule comprising only 2 doses as opposed to 3 or 4 doses, once VTs have been controlled, is an attractive policy option [6]. In 2020, the United Kingdom became the first country to move to a 2-dose (1 + 1) schedule for 13-valent PCV (PCV13) in the childhood vaccination program [7]. However, evidence to support the use of 1 + 1 schedules in settings with low or heterogeneous vaccine coverage is not available.

Conceptually, the population-level indirect effects of vaccination programs are achieved through sufficient vaccination coverage among age groups primarily responsible for

transmission [8,9]. With sufficient indirect effects, near elimination of VT transmission and disease can be achieved. Near elimination of VT disease has largely occurred in settings that have rapidly reached very high rates of coverage; hence, it is often assumed that high PCV coverage is required to achieve elimination, but there is limited evidence for this [10]. Settings with lower rates of coverage often lack the disease surveillance required to demonstrate the indirect effects of vaccines. It is therefore not clear whether VT elimination is possible for nearly half of the countries that have introduced PCV, which have less than 90% PCV coverage [11]. To date, most studies evaluating indirect effects from PCV have used aggregate data and ecological study designs, such as before-and-after or time-series analyses. Few have incorporated PCV coverage into their analyses.

Australia is a geographically diverse setting with a combination of dense urban populations and sparsely populated rural areas. Within Australia, Aboriginal and Torres Strait Islander children (henceforth referred to as Indigenous children) experience significantly higher rates of morbidity across a range of infectious diseases including pneumococcal disease, likely related to higher levels of social disadvantage [12,13]. Pre-PCV introduction, IPD rates in Indigenous children from Central Australia were some of the highest in the world [14].

In June 2001, 7-valent PCV (PCV7) was introduced for high-risk children, including Indigenous children and children with specified medical conditions, at ages 2, 4, and 6 months. Children who were medically at risk received a fourth dose of PCV7 at 12 months and a dose of 23-valent pneumococcal polysaccharide vaccine (PPV23) at 5 years of age, while Indigenous children in high-incidence jurisdictions received PPV23 at 18 to 24 months of age. A 3-dose PCV program (2, 4, and 6 months) became universal in January 2005, with catch-up doses for children aged less than 2 years of age. In July 2011, PCV13 replaced PCV7 with a catch-up program funding a supplemental dose of PCV13 for children 12 to 35 months of age [15]. For Indigenous children in high-incidence jurisdictions, a fourth dose of PCV13 replaced the PPV23 vaccine at 18 to 23 months.

Using the same large linked immunisation and hospitalisation dataset, indirect protection following PCV introduction previously in Australia has been reported previously at the state level for children less than 2 years of age, register with all-cause pneumonia declining among unvaccinated Indigenous (12% reduction; 95% confidence interval [CI] 3 to 25) and non-Indigenous children (45% reduction; 95% CI 41 to 49) [16].

In this study, we extended this analysis by examining the association between PCV coverage in small geographic units within 2 states for both incidence of IPD and hospitalised pneumonia among undervaccinated children, in periods when coverage was changing rapidly. Our study provides a unique opportunity to evaluate indirect protection against pneumococcal disease in the absence of the widespread use of the booster dose, making this analysis more applicable to many low-income countries where, at the time of the study, the 3 + 0 schedule is primarily used [17].

## Methods

### Study design

Our study design was a retrospective population-based cohort that used a subset of children under 5 years of age from a dataset of all children born in New South Wales (NSW) and Western Australia (WA) between 2001 and 2012, a combined birth cohort of approximately 1.37 million [18]. Birth records were probabilistically linked (using name, date of birth, residential address, and sex) to health data including vaccination register, IPD notification data, and hospitalisation data [18].

## Linked data sources

Vaccination status was obtained from the Australian Childhood Immunisation Register (ACIR), which includes all children enrolled in the publicly funded healthcare system and comprises approximately 99% of children in Australia by age 12 months [19]. IPD cases, defined as isolation of *S. pneumoniae* by culture or detection of nucleic acid from a normally sterile site, are notified as part of state-based passive surveillance systems [19]. Hospitalisation data covered all inpatient separations (discharges, transfers, and deaths) and included primary diagnosis and up to 50 (NSW) or 20 (WA) secondary diagnoses (coded using the Australian version of the International Classification of Diseases [ICD-AM] coding system). IPD data were available from January 2001 onwards, while hospitalisation data were available from July 2001 onwards. Demographic and health risk factor data were obtained from state perinatal data collection and birth registries (Table A in S1 File) [19]. See Fig A in S1 File for a flow chart of the study cohort and data sources.

## Study definitions

PCV7-type IPD was defined as IPD due to serotypes in PCV7, i.e., serotypes 4, 6B, 9V, 14, 18C, 19F, and 23F. PCV13, non-PCV7-type IPD was defined as IPD due to the additional 6 serotypes in PCV13, i.e., serotypes 1, 3, 5, 6A, 7F, and 19A. For all-cause pneumonia, we identified all hospitalisations with a pneumonia-related diagnostic code in the principal or additional diagnosis fields. Pneumococcal or lobar pneumonia hospitalisations were restricted to hospitalisations coded as either pneumococcal pneumonia (J13) or lobar pneumonia (J18.1) (Table B in S1 File). Interhospital transfers and admissions within 14 days of a previous separation date were merged and classified as a single episode as per previous analyses [20]. All outcomes were evaluated for children under 5 years of age.

We defined a child as vaccinated with PCV if they received an adequate number of doses to develop a protective immune response against vaccine serotypes at least 14 days prior to onset of any study outcomes, i.e., 2 or more PCV doses administered at less than 12 months of age, or at least 1 PCV dose administered after the age of 12 months [21]. Otherwise, cases were classified as undervaccinated. Given children who have received 1 dose of PCV at less than 12 months of age may have partial protection, we have also completed sensitivity analyses examining indirect effects among completely unvaccinated children.

## Statistical analyses

For descriptive analyses of PCV coverage over time, we calculated and graphed coverage among children less than 5 years of age at 3-monthly intervals (31 March, 30 June, 30 September, and 31 December) from 2001 to 2012. For example, PCV coverage among children less than 5 years of age at the final quarter of 2012 (assessed at 31 December) would be calculated for those born between 31 December 2007 and 30 December 2012. Coverage was calculated among children under 5 years of age to aid comparison with previous studies [22,23] and because previous research suggests that vaccination of children up to 5 years of age are key for the generation of indirect effects [22,24].

To visualise any geographic heterogeneity in PCV coverage, PCV coverage was mapped within statistical local areas (SLAs) among children under 5 years of age at 3 time points—December 2005, December 2009, and December 2012. The 3 time points were chosen to represent the early post-PCV7 period, the late post-PCV7 period, and the early PCV13 period. SLAs are the second smallest geographical spatial unit used by the Australian Bureau of Statistics [25]. In NSW and WA, there are 199 and 155 SLAs, respectively. SLAs vary in population size from less than 100 children in some rural areas to around 11,000 children under 5 years of age

in urban areas. SLAs with fewer than 100 children under 5 years of age were excluded to enable meaningful estimates of PCV coverage for the different age groups.

Crude rates of IPD and pneumonia hospitalisation among all children and undervaccinated children under 5 years of age over 3 time periods were calculated: targeted PCV program (2001 to 2004), universal PCV7 program (2005 to 2010), and universal PCV13 program (2011 to 2012). Person-time for each child started at birth and was censored at the earliest of the following: death, when the child reached 5 years of age or at the end of the study period. For the undervaccinated group, person-time was also censored when the child was considered vaccinated as previously defined.

To determine the association between PCV coverage and indirect effects against IPD and hospitalised pneumonia (primary analysis), we first calculated the incidence of each disease outcome during the person-time for which children were undervaccinated up to 5 years of age. We divided each child's undervaccinated person-time into successive 3-month intervals and linked the intervals to PCV coverage in their SLA of residence at that same time period. SLA of residence was based on mother's residence at child's birth. Using Poisson regression models, we examined the association between SLA-level PCV coverage and disease incidence in undervaccinated children. We used robust variance estimates to account for clustering, accommodating multiple coverage estimates for each individual and checked linearity assumptions by plotting the log of the disease rates against PCV coverage.

Analyses relating to PCV7-type IPD used the full study period from January 2001 and December 2012, while analyses relating to PCV13, non-PCV7-type IPD were restricted to from January 2009 onwards to 18 months prior to PCV13 introduction. Analyses relating to pneumonia hospitalisations used data from July 2001 onwards due to lack of availability of hospitalisation data in NSW prior to this date.

Potential confounding variables were selected a priori using a directed acyclic graph (DAG), informed by relevant literature, and refined through expert consultation (S1 File, pp 6). These were age group, calendar year, Indigenous status, season, Index of Relative Socioeconomic Advantage and Disadvantage (IRSAD) score, Accessibility or Remoteness Index of Australia (ARIA) category, birth weight, gestational age, maternal smoking during pregnancy, number of previous pregnancies, and previous hospitalisation with an ICD code corresponding to the presence of a medical condition increasing risk of IPD.

We graphed predicted incidence rates of each outcome among undervaccinated children under 5 at each decile of PCV coverage, accounting for the balance of covariates across all the individuals [26]. We also calculated the estimated population preventable fraction of PCV7-type IPD and all-cause pneumonia hospitalisations for each decile of PCV coverage, defined as the proportion of disease among undervaccinated children estimated to be preventable by increasing vaccine coverage among different age groups [27].

We conducted subgroup analyses, examining PCV7-type IPD and all-cause pneumonia hospitalisations among (1) children too young to be vaccinated (under 4 months of age); (2) undervaccinated children in urban and rural Australia; and (3) undervaccinated Indigenous children. We expected that associations for these subgroups may vary as the relationship between PCV coverage and pneumococcal disease is modulated by the dynamics of pneumococcal transmission, which, in turn, differs with population density, age group, risk factors, and household structure [13,28].

This study is reported as per the REporting of studies Conducted using Observational Routinely-collected Data (RECORD) guideline (S1 RECORD Checklist). Statistical analyses were performed according to a prospective data analysis plan (see S1 Text).

Ethical approval was obtained from the following: The Aboriginal Health and Medical Research Council Ethics Committee (approval ID: 931/13), AIHW Ethics Committee

(approval ID: EC 2012/4/62), Department of Health Human Research Ethics Committee (approval ID: 1/2013), Department of Health–WA Human Research Ethics Committee (approval ID: 2012/75), NSW Population and Health Service Research Ethics Committee (approval ID: HREC/13/CIPHS/15), and Western Australian Aboriginal Health Ethics Committee (approval ID: 459).

## Results

### Study cohort and outcomes

The cohort comprised 1,365,893 singleton children born between January 2001 and December 2012, of which 1,223,803 lived in urban areas, 106,470 lived in rural areas, and 66,484 were Indigenous [16,29]. The characteristics of this cohort are described in Table 1. During 5,852,233 person-years of follow-up among children under 5 years of age, 1,427 cases of IPD were notified: 43.2% due to PCV7 serotypes, 28.4% due to PCV13–non-PCV7 serotypes, 17.7% due to non-PCV serotypes, and 10.7% lacked serotype data. Over the same period, 34,757 children experienced at least 1 episode of pneumonia hospitalisation, of which 799 (2.3%) were coded as pneumococcal or lobar pneumonia. Children were undervaccinated against any PCV during 1,823,401 person-years of follow-up (31.2% of total person-years), with 83% of this time in children who had received no PCV doses.

### Incidence of invasive pneumococcal disease (IPD) and pneumonia hospitalisations by period

Among undervaccinated children, rates of all-cause IPD decreased from 73.3 in the targeted PCV7 period to 17.1 per 100,000 person-years in the universal PCV13 period (Table 2). Rates of PCV13, non-PCV7-type IPD increased in the universal PCV7 period, before decreasing in the universal PCV13 period (Table 2). In the universal PCV7 period, rates of PCV7-type IPD were higher in undervaccinated children compared to all children. Comparable trends were observed among different age groups and among both Indigenous and non-Indigenous children (Table D in S1 File). Annual rates of PCV7 and PCV13, non PCV7-type IPD, are presented alongside annual PCV coverage estimates in Tables E and F in S1 File.

Among undervaccinated children, rates of all-cause pneumonia decreased from 90.7 to 50.3 per 10,000 person-years from targeted PCV7 to universal PCV13 periods (Table 3). Substantial reductions were observed for both all-cause pneumonia and pneumococcal or lobar pneumonia across different age groups by vaccination period (Table G in S1 File).

### PCV coverage

Among children under 5 years, coverage of any PCV increased steeply to 40% reflecting a catch-up program up to 2 years of age, then more gradually over the next 4 years as vaccinated children aged into older age groups, reaching 86% by December 2012 (Fig 1).

Figs 2 and 3 depict the geographical heterogeneity in PCV coverage at the SLA level among children aged 12 to 23 months at the end of each of the following years: 2005, 2009, and 2012, with greatest heterogeneity in the year of universal PCV introduction (2005). The median and interquartile ranges of SLA-level PCV coverage by year are presented in Tables E and F in S1 File.

### PCV coverage and indirect protection against vaccine-type IPD

Fig 4 shows the association between PCV coverage and incidence of PCV7-type IPD for undervaccinated children aged under 5 years. The number of IPD cases and the

**Table 1. Cohort characteristics by Indigenous status, New South Wales and Western Australia, 2001–2012.**

| | | Non-Indigenous, n (%), N = 1,299,409 | Indigenous, n (%), N = 66,484 |
|---|---|---|---|
| State of birth registration | New South Wales | 993,936 (76.5) | 45,551 (68.5) |
| | Western Australia | 305,473 (25.5) | 20,933 (31.5) |
| Sex | Male | 668,021 (51.4) | 34,284 (51.6) |
| | Female | 631,388 (48.6) | 32,200 (48.4) |
| Socioeconomic index | 91%–100% (least disadvantaged) | 160,519 (12.4) | 990 (1.5) |
| | 76%–90% | 199,055 (15.3) | 2,739 (4.1) |
| | 26%–75% | 582,833 (44.9) | 21,704 (32.7) |
| | 11%–75% | 196,074 (15.1) | 15,371 (23.1) |
| | 0%–10% (most disadvantaged) | 126,124 (9.7) | 21,996 (33.1) |
| | Missing | 34,804 (2.7) | 3,684 (5.5) |
| Remoteness index | Major cities (urban) | 989,470 (76.2) | 27,633 (42.6) |
| | Inner regional (urban) | 189,454 (14.6) | 17,246 (25.9) |
| | Outer regional (rural) | 71,056 (5.5) | 10,899 (16.4) |
| | Remote (rural) | 13,681 (1.1) | 4,813 (7.2) |
| | Very remote (rural) | 3,310 (0.3) | 2,711 (4.8) |
| | Missing | 32,438 (2.5) | 3,182 (4.8) |
| Number of previous pregnancies (parity) | 0 | 520,276 (40.0) | 22,119 (33.3) |
| | 1 | 434,073 (33.4) | 16,512 (24.8) |
| | 2 | 205,396 (15.8) | 11,002 (16.6) |
| | ≥3 | 138,257 (10.6) | 16,786 (25.3) |
| Birthweight (g) | <1,500 | 7,105 (0.6) | 833 (1.3) |
| | 1,500–2,499 | 45,846 (3.5) | 5,417 (8.2) |
| | 2,500–3,499 | 668,358 (51.4) | 36,029 (54.2) |
| | 3,500–4,499 | 555,005 (42.7) | 23,126 (34.8) |
| | ≥4,500 | 22,771 (1.8) | 1,065 (1.6) |
| | Missing | 324 (0.0) | 14 (0.0) |
| Prematurity (gestational age <33 weeks) | Preterm | 8,539 (0.7) | 993 (1.5) |
| | Missing | 156 (0.0) | 30 (0.1) |
| Parental smoking | Yes | 151,309 (11.6) | 30,309 (45.6) |
| | No | 1,145,210 (88.1) | 36,094 (54.2) |
| | Missing | 2,890 (0.2) | 81 (0.1) |
| Hospital admission aged <6 weeks | | 193,020 (14.9) | 13,566 (20.4) |
| Any hospital admission for risk condition | | 47,705 (3.7) | 3,382 (5.1) |

undervaccinated person-time available for analysis at different levels of PCV coverage are provided in Table H in S1 File.

For each percentage point increase in PCV coverage among children 0 to 59 months of age, the adjusted incidence of PCV7-type IPD decreased by 3.3% (95% CI 2·5 to 4·2, <0.001) among undervaccinated children under 5 years of age (Table 4). The steepest declines in PCV7-type IPD occurred between 0% and 50% coverage among children 0 to 59 months of age (Fig 4).

Additionally, incidence of PCV13-type IPD among undervaccinated children decreased as PCV13 coverage increased; however, CIs crossed the null value (adjusted incidence rate ratio [aIRR] 0.989; 95% CI 0.971 to 1.007; $p = 0.241$) (Fig 4 and Table 4).

Subgroup analyses among children less than 4 months of age, children living in urban areas, children living in rural areas, and Indigenous children showed similar trends, although CIs for Indigenous and rural subgroup analyses were wide as sample sizes were small (see Fig 5

**Table 2. Rates of IPD among all children and undervaccinated children\* by vaccination period, New South Wales and Western Australia, 2001–2012.**

| | All children | | Undervaccinated children (PCV13) | |
|---|---|---|---|---|
| | Cases/person-years | IPD rate (95% CI)[†] | Cases/person-years | IPD rate[†] |
| **All-cause IPD** | | | | |
| Targeted PCV7 (2001–2004) | 603/835,128 | 7.2 (6.7–7.8) | 595/811,740 | 7.3 (6.7–7.9) |
| Universal PCV7 (2005–2010) | 595/3,275,465 | 1.8 (1.7–2.0) | 181/944,751 | 1.9 (1.7–2.2) |
| Universal PCV13 (2011–2012) | 225/1,741,640 | 1.3 (1.2–1.5) | 30/175,044 | 1.7 (1.2–2.5) |
| **PCV7-type IPD** | | | | |
| Targeted PCV7 (2001–2004) | 460/835,333 | 5.5 (5.0–6.0) | 455/811,919 | 5.6 (5.1–6.1) |
| Universal PCV7 (2005–2010) | 144/3,276,676 | 0.4 (0.4–0.5) | 103/945,020 | 1.1 (0.9–1.3) |
| Universal PCV13 (2011–2012) | 13/1,742,423 | 0.1 (0.0–0.1) | 3/175,090 | 0.2 (0.1–0.5) |
| **PCV13, non-PCV7-type IPD** | | | | |
| Targeted PCV7 (2001–2004) | 29/833,868 | 0.4 (0.2–0.5) | 29/835,868 | 0.4 (0.2–0.5) |
| Universal PCV7 (2005–2010) | 265/3,277,575 | 0.8 (0.7–0.9) | 265/3,277,444 | 0.8 (0.7–0.9) |
| Universal PCV13 (2011–2012) | 111/1,741,984 | 0.6 (0.5–0.8) | 101/1,282,026 | 0.8 (0.7–1.0) |

\*Undervaccinated for both PCV7 and PCV13.

[†]Per 10,000 person-years.

IPD, invasive pneumococcal disease; PCV7, 7-valent PCV; PCV13, 13-valent PCV.

and Table I in S1 File). Sensitivity analyses among completely unvaccinated children yielded similar results (Tables J and K in S1 File).

Table 5 shows the estimated preventable fraction of PCV7-type IPD in undervaccinated children less than 5 years of age for each decile increase in PCV coverage. Vaccinating half of children less than 5 years of age is estimated to prevent 81.4% (95% CI 71.2 to 88.0) of PCV7-type IPD, while 90% coverage prevents 95.2% (95% CI 89.4 to 97.8).

## PCV coverage and indirect protection against pneumonia hospitalisations

Similarly, there were inverse associations between PCV coverage and incidence of both all-cause pneumonia and pneumococcal or lobar pneumonia hospitalisations among undervaccinated children (Fig 6 and Table 6). The number of pneumonia hospitalisations and

**Table 3. Rates of pneumonia hospitalisations among all children and undervaccinated children\*, New South Wales and Western Australia by vaccination period, 2001–2012.**

| | All children | | Undervaccinated children (PCV13) | |
|---|---|---|---|---|
| | Cases/person-years | Hospitalisation rate[†] (CI) | Cases/person-years | Hospitalisation rate[†] (CI) |
| **All-cause pneumonia hospitalisations** | | | | |
| Targeted PCV7 (2001–2004) | 7,704/811,760 | 94.9 (92.8–97.0) | 7,156/788,957 | 90.7 (88.6–92.8) |
| Universal PCV7 (2005–2010) | 18,026/3,229,066 | 55.8 (55.0–56.6) | 4,589/931,337 | 49.3 (47.9–50.7) |
| Universal PCV13 (2011–2012) | 9,009/1,720,705 | 52.4 (51.3–53.4) | 876/174,256 | 50.3 (47.0–53.7) |
| **Pneumococcal or lobar pneumonia hospitalisations** | | | | |
| Targeted PCV7 (2001–2004) | 284/820,697 | 3.5 (3.1–3.9) | 266/812,094 | 3.3 (2.9–3.7) |
| Universal PCV7 (2005–2010) | 364/3,275,939 | 1.1 (1.0–1.2) | 100/945,250 | 1.1 (0.9–1.3) |
| Universal PCV13 (2011–2012) | 158/1,744,642 | 0.9 (0.8–1.1) | 18/175,070 | 1.0 (0.6–1.6) |

\*Undervaccinated for both PCV7 and PCV13.

[†]Per 10,000 person-years.

PCV7, 7-valent PCV; PCV13, 13-valent PCV.

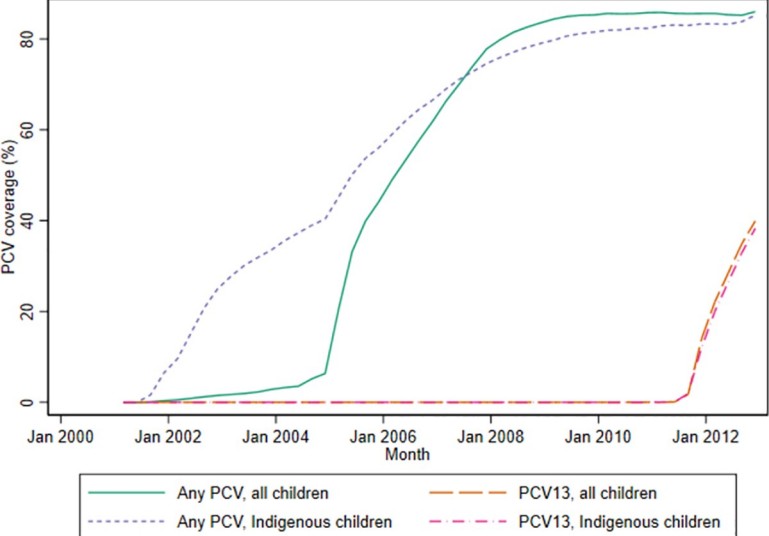

**Fig 1. Quarterly coverage* of any PCV and PCV13 among children under 5 years of age, New South Wales and Western Australia, 2001–2012.** *Vaccinated defined as 2 or more PCV doses administered at <12 months of age or at least 1 PCV dose administered after the age of 12 months. PCV, pneumococcal conjugate vaccine; PCV13, 13-valent PCV; WA, Western Australia.

undervaccinated person-time at the different levels of PCV coverage are provided in Table L in S1 File. For each percentage point increase in PCV coverage among children less than 5 years of age, the adjusted incidence of pneumococcal or lobar pneumonia hospitalisations decreased by approximately 1.5% (95% CI 0.7 to 2.6; *p* = 0.001) (Table 6). For each percentage

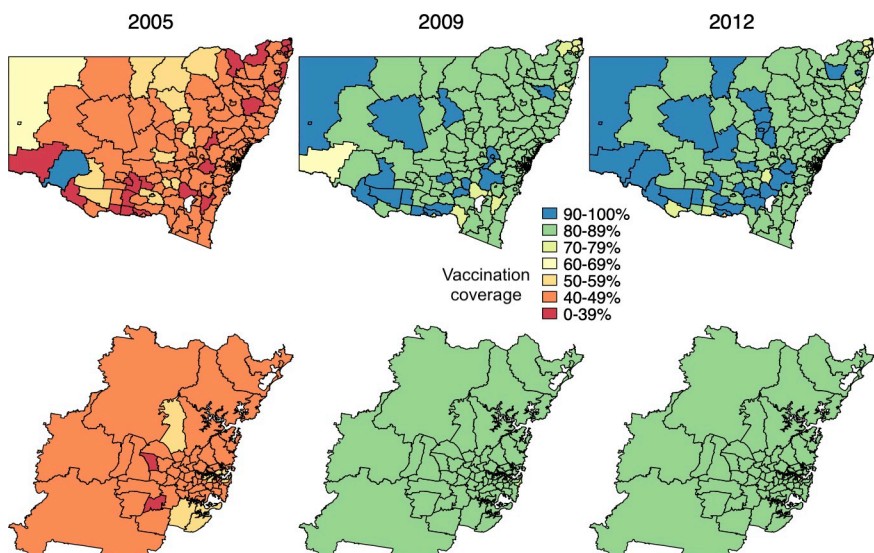

**Fig 2. Coverage* of any PCV among children less than 5 years of age by SLA and year, New South Wales, 2001–2012; top row: New South Wales (state), bottom row: Sydney (capital city); SLAs coloured white have been excluded due to small numbers of children.** *Vaccinated defined as 2 or more PCV doses administered at <12 months of age or at least 1 PCV dose administered after the age of 12 months. SLA digital boundaries (ASGC 2006) for this map were obtained from the Australian Bureau of Statistics (https://www.abs.gov.au/) CC BY 4.0. PCV, pneumococcal conjugate vaccine; SLA, statistical local area.

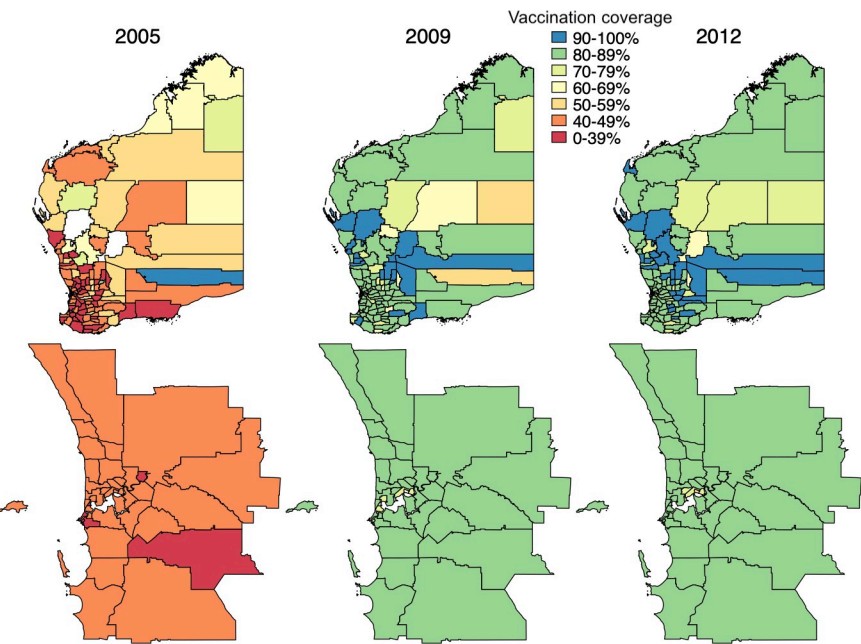

**Fig 3. Coverage* of any PCV among children 12–23 months of age by SLA and year, Western Australia, 2001–2012; top row: Western Australia (state), bottom row: Perth (capital city); SLAs coloured white have been excluded due to small numbers of children.** *Vaccinated defined as 2 or more PCV doses administered at <12 months of age or at least 1 PCV dose administered after the age of 12 months. SLA digital boundaries (ASGC 2006) for this map were obtained from the Australian Bureau of Statistics (https://www.abs.gov.au/) CC BY 4.0. PCV, pneumococcal conjugate vaccine; SLA, statistical local area.

point increase in PCV coverage among children 0 to 59 months of age, the adjusted incidence of all-cause pneumonia hospitalisations decreased by approximately 0.9% (95% CI 0.6 to 1.0; $p < 0.001$) (Table 6).

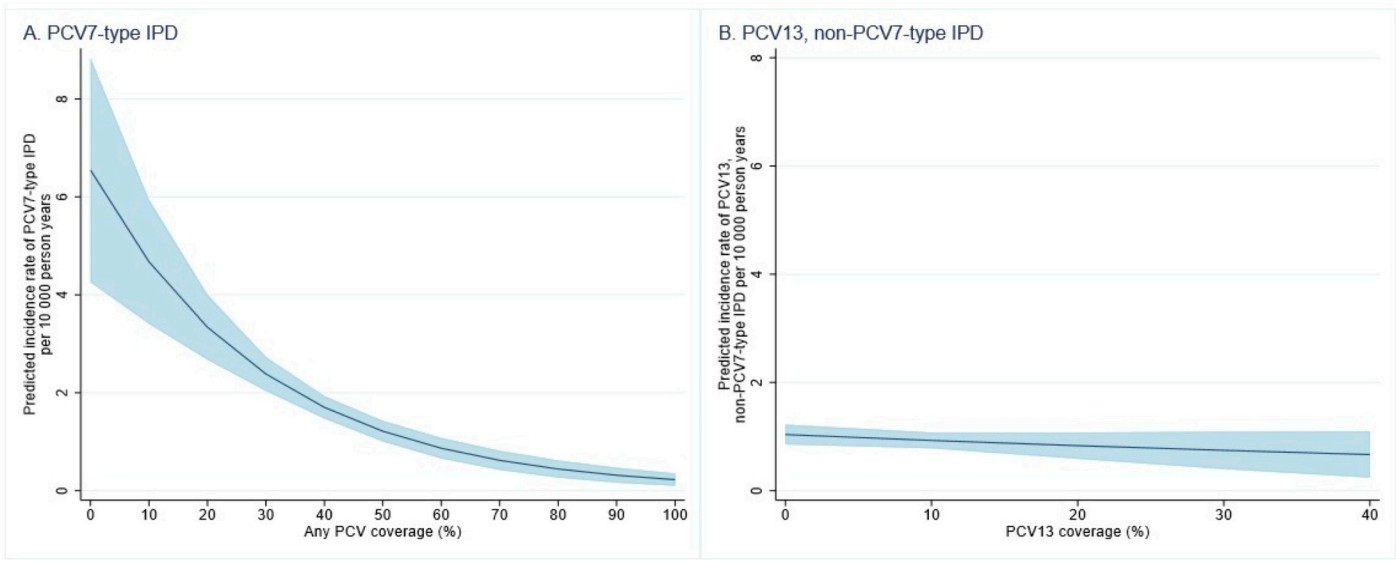

**Fig 4. Estimated rate of VT IPD incidence* and 95% CIs (blue shading) among undervaccinated children under 5 years, at each decile of PCV coverage, New South Wales and Western Australia, 2001–2012.** *PCV13, non-PCV7-type IPD incidence was only estimated up to 60% as there were few data points available. CI, confidence interval; IPD, invasive pneumococcal disease; PCV, pneumococcal conjugate vaccine; PCV7, 7-valent PCV; PCV13, 13-valent PCV; VT, vaccine-type.

**Table 4. Crude and adjusted\* incidence rate ratios of PCV7-type IPD among undervaccinated children under 5 years, by percent increase in PCV coverage† among children under 5 years of age, New South Wales and Western Australia, 2009–2012.**

|  | Crude | | Adjusted\* | |
|---|---|---|---|---|
|  | Incidence rate ratio (95% CI) | *p*-value | Incidence rate ratio (95% CI) | *p*-value |
| PCV7-type IPD (*n* = 8,579,771§) | 0.965 (0.961–0.969) | <0.001 | 0.967 (0.958–0.975) | <0.001 |
| PCV13, non-PCV7-type IPD (*n* = 8,941,562§) | 0.985 (0.972–0.998) | 0.029 | 0.989 (0.971–1.007) | 0.241 |

\*Adjusted by Indigenous status, season, IRSAD score, ARIA category, birth weight, gestational age, maternal smoking during pregnancy, number of previous pregnancies, and previous hospitalisation for a range of high-risk medical comorbidities.

†Coverage of any PCV—inclusive of both PCV7 and PCV13.

§Number of observations for adjusted analysis (quarterly person-time).

ARIA, Accessibility or Remoteness Index of Australia; CI, confidence interval; IPD, invasive pneumococcal disease; IRSAD, Index of Relative Socio-economic Advantage and Disadvantage; PCV, pneumococcal conjugate vaccine; PCV7, 7-valent PCV; PCV13, 13-valent PCV.

Subgroup analyses for all-cause pneumonia hospitalisations among children living in urban areas, children living in rural areas, and Indigenous children showed similar trends, although, again, CIs for Indigenous and rural subgroup analyses were wide as there were fewer cases and individuals in these subgroups (Table M in S1 File).

Table 7 shows the preventable fraction of all-cause pneumonia hospitalisations in undervaccinated children less than 5 years of age by each decile increase in PCV coverage. We estimate that vaccinating 50% of children less than 5 years of age will prevent 33.3% (95% CI 27.3 to 38.8) of all-cause pneumonia hospitalisations in undervaccinated children less than 5 years, while vaccinating 90% prevents 51.7% (95% CI 43.7 to 58.6).

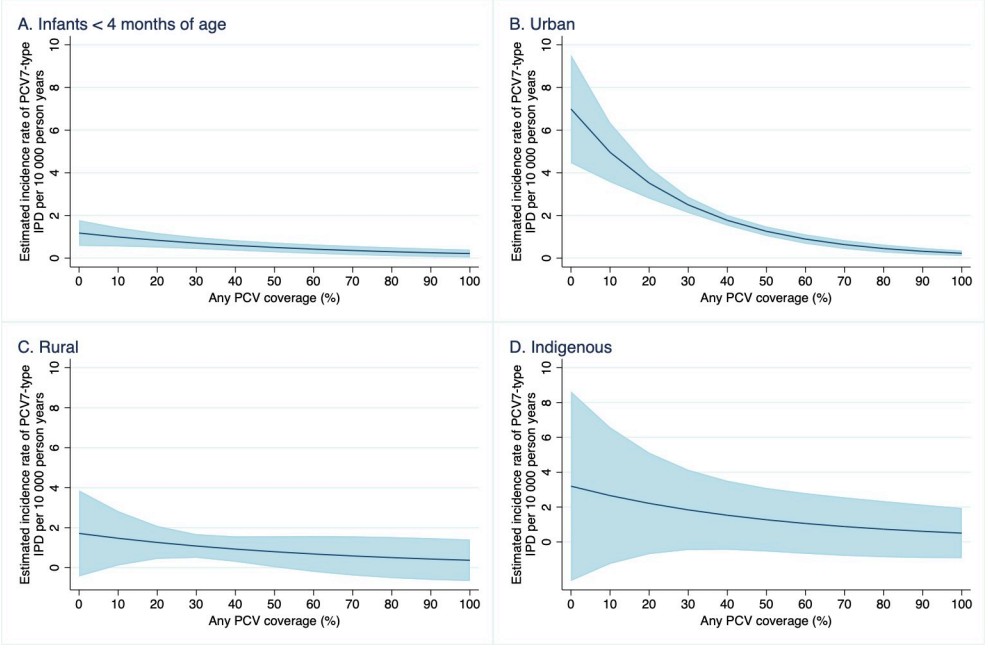

**Fig 5. Estimated rate of VT IPD incidence\* and 95% CIs (blue shading) among 3 subgroups of undervaccinated children under 5 years (urban, rural, and Indigenous), at each decile of PCV coverage, New South Wales and Western Australia, 2001–2012.** CI, confidence interval; IPD, invasive pneumococcal disease; PCV, pneumococcal conjugate vaccine; PCV7, 7-valent PCV; VT, vaccine-type.

**Table 5. Preventable fraction of PCV7-type IPD in undervaccinated children under 5 years at deciles of PCV coverage, New South Wales and Western Australia, 2001–2012.**

| Coverage (%) | Preventable fraction of PCV7-type IPD at deciles of PCV coverage |
|---|---|
| 10 | 28.6 (22.1–34.6) |
| 20 | 49.0 (39.2–57.2) |
| 30 | 63.6 (52.6–72.0) |
| 40 | 74.0 (63.1–81.7) |
| 50 | 81.4 (71.2–88.0) |
| 60 | 86.7 (77.6–92.2) |
| 70 | 90.5 (82.5–94.9) |
| 80 | 93.2 (86.4–96.6) |
| 90 | 95.2 (89.4–97.8) |

IPD, invasive pneumococcal disease; PCV, pneumococcal conjugate vaccine; PCV7, 7-valent PCV.

## Discussion

The findings from our novel analyses indicate that relatively low levels of coverage are required to generate indirect protection in children against IPD and pneumonia hospitalisations in Australia. These results challenge previous understandings that high coverage is required for substantial indirect protection. Consistent with previous studies, our estimates indicate an inverse relationship between PCV coverage and indirect effects on VT IPD [30]. A meta-analysis of PCV impact studies reported a 0.5% reduction in IPD (95% credible interval 0.2% to 1.1%) among children and adults for each percentage increase in vaccine coverage among children, accounting for country-level confounders [31]. However, in contrast to previous pneumococcal carriage studies, which reported statistically significant indirect effects at 58% to 75% coverage, we estimated substantial indirect effects at lower rates of PCV coverage [22,23,32]. While not directly comparable, the indirect effects of PCV against disease can be expected to reflect indirect effects on carriage, since these effects are mediated by reductions in carriage [24]. The variation in results may be due to differences in analytic method. Compared

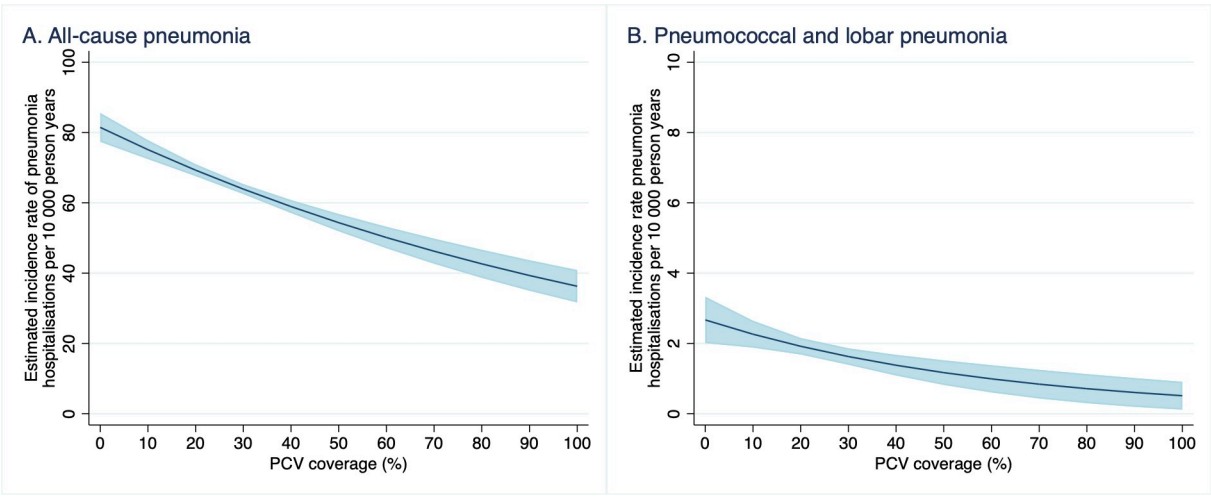

**Fig 6. Estimated rate of pneumonia hospitalisation and 95% CIs (blue shading) among undervaccinated children at each decile of PCV coverage, New South Wales and South Australia, 2001–2012.** CI, confidence interval; PCV, pneumococcal conjugate vaccine.

**Table 6. Crude and adjusted\* incidence rate ratios of pneumonia hospitalisations among undervaccinated children under 5 years, by percent increase in PCV coverage[†] among children under 5 years of age, New South Wales and Western Australia, 2009–2012.**

| | Crude | | Adjusted | |
|---|---|---|---|---|
| | Incidence rate ratio (95% CI) | *p*-value | Incidence rate ratio (95% CI) | *p*-value |
| Pneumococcal or lobar pneumonia hospitalisation (*n* = 6,885,749[§]) | 0.980 (0.976–0.984) | <0.001 | 0.985 (0.974–0.993) | 0.001 |
| All-cause pneumonia hospitalisation (*n* = 6,794,370[§]) | 0.993 (0.992–0.994) | <0.001 | 0.991 (0.990–0.994) | <0.001 |

\*Adjusted by Indigenous status, age group, season, IRSAD score, ARIA category, birth weight, gestational age, maternal smoking during pregnancy, number of previous pregnancies, and previous hospitalisation for a range of high-risk medical comorbidities.

[†]Any PCV—inclusive of both PCV7 and PCV13.

[§]Number of observations for adjusted analysis (quarterly person-time).

ARIA, Accessibility or Remoteness Index of Australia; CI, confidence interval; IRSAD, Index of Relative Socio-economic Advantage and Disadvantage; PCV, pneumococcal conjugate vaccine.

to annual carriage surveys conducted by Hammitt and colleagues [32], we analysed at quarterly intervals allowing us to detect indirect effects at lower coverage rates. Grant and colleagues defined indirect effects as a statistically significant reduction in VT carriage prevalence, a measure that depends on both the effect size and sample size [23]. While Loughlin and colleagues chose to define indirect effects a priori as a 50% decline in VT carriage [22]. By modelling the association, we were able to quantify the estimated degree of indirect effects at varying coverage rates, facilitating future comparisons between studies with varying sample sizes and methods. Differences in the relationship between PCV coverage and indirect effects between studies may also be due to differences in the vaccine type studied (i.e., PCV7, PCV10, or PCV13) or context.

Importantly, we found similar trends among rural populations and Indigenous populations, although these effects were smaller and had large confidence intervals, which likely reflects the smaller sample size available for analysis. These findings are important as they reflect the estimated indirect effects in different settings with different patterns of social mixing and therefore patterns of pneumococcal disease transmission. The modelled incidence of PCV7-type disease at zero coverage was lower than expected, given Indigenous children are known to be at high risk of pneumococcal disease. This is likely due to the earlier introduction of PCV for Indigenous children in 2001, which meant undervaccinated Indigenous children were already benefiting from some degree of indirect effects at the start of our study's

**Table 7. Preventable fraction of all-cause pneumonia hospitalisations in undervaccinated children under 5 years at deciles of PCV coverage, New South Wales and Western Australia, 2001–2012.**

| Coverage (%) | Preventable fraction of all-cause pneumonia hospitalisation at deciles of PCV coverage |
|---|---|
| 10 | 7.7 (6.2–9.3) |
| 20 | 14.9 (11.9–17.8) |
| 30 | 21.6 (17.4–25.5) |
| 40 | 27.6 (22.5–32.4) |
| 50 | 33.3 (27.3–38.8) |
| 60 | 38.4 (31.8–44.5) |
| 70 | 43.2 (36.0–49.7) |
| 80 | 47.7 (40.0–54.4) |
| 90 | 51.7 (43.7–58.6) |

PCV, pneumococcal conjugate vaccine.

observation period. Without sufficient baseline data prior to vaccine introduction, our analyses likely underestimate the full extent of indirect effects among Indigenous children. Given that Indigenous children make up 17.3% of our rural cohort, the earlier introduction of PCV among Indigenous children may also have impacted on our estimates of indirect effects in rural settings. However, the low baseline rates of disease may also be attributable to a more sparsely populated rural population with less social mixing compared to urban settings [33]. Previous studies on the direct effect of PCV found the vaccine to be equally effective among Indigenous children and children from rural areas, suggesting that our results are not due to reduced vaccine effectiveness in these groups. Evidence of indirect effects among children under 4 months of age, albeit not statistically significant, is also important to note since these children are too young to be fully vaccinated across the time period while vaccine eligibility and coverage change for other age groups.

Our analyses also indicated that increasing indirect protection against pneumonia hospitalisations was associated with rising PCV coverage. Pneumonia comprises a substantially larger disease burden than IPD and is a crucial determinant of vaccine cost-effectiveness of a PCV program. Our findings are consistent with previous analysis using the same dataset, which found 38% and 28% reductions in all-cause pneumonia hospitalisations among children <2 and 2 to 4 years of age, respectively, 2 years post-PCV7 introduction in Australia [34]. Similar relationships have been reported for PCV uptake and pneumonia hospitalisations in Brazil [30]. While our analyses, accounted for individual-level risk factors for pneumonia, it did not account for temporal trends, including changes admission practices or pneumonia epidemiology, which could account for an overall decline in pneumonia across this time period. This may explain the unexpectedly large estimate that over 50% of all-cause pneumonia among undervaccinated children is preventable at 90% PCV coverage among children under 5.

Our analyses did not identify a clear threshold at which PCV7-type IPD was eliminated—with 5% of PCV7-type remaining at 90% coverage. Similarly, a prior study of the long-term impacts of PCV against IPD in Australia reported reductions of 98% and 96% in VT IPD among children <2 and 2 to 4 years old 8 years after vaccine introduction [3]. Our results may reflect the inherent limitations of the vaccine or schedule or insufficient period of monitoring post-PCV introduction to see full vaccine impacts; alternatively, there may be a lag in observing the full effects of indirect protection after reaching a certain PCV coverage [8], which was not accounted for in this model.

While our analyses of PCV7-type IPD showed the steep declines in disease, especially up to 50% PCV coverage, we observed flatter associations between PCV coverage and both PCV13-type IPD and all-cause pneumonia hospitalisations. These results are consistent with previous studies indicating that higher coverage rates (58% to 75%) were required for substantial indirect effects using PCV10 and PCV13. This may be explained by previous research indicating lower vaccine effectiveness against the additional serotypes in PCV13 [35,36]. However, there is a high degree of uncertainty around our PCV13 estimates since we have limited data post-PCV13 introduction (1.5 years) and a small number of PCV13, non-PCV7 type IPD cases. At the end of our study period, PCV13 coverage had only reached a median of 32.1% among children under 5. Therefore, our analyses are unable to capture the full indirect effects of PCV13, which are likely to increase as vaccine coverage rises over time. Smaller effect sizes for the estimated indirect vaccine effects against all-cause pneumonia hospitalisations may be due to the nonspecific nature of the outcome and/or differences in vaccine efficacy against the different spectrum of serotypes causing pneumonia, compared to IPD [37].

The availability of a large population-based linked dataset and analyses at fine spatial and temporal resolutions enabled precise evaluation of indirect effects in Australia; however, there are some limitations. While our cohort is estimated to represent 97.5% of live births, it does

not include children who have migrated into the 2 states or children who were never registered at birth. SLA of residence was based on the mother's SLA of residence at the time of the child's birth, and we do not have updated data on SLA of residence if a family or child moved. We acknowledge that SLAs will not perfectly capture the boundaries of communities that interact socially, in which to measure indirect protection. As with all observational studies, causation cannot be assured. However, we observed a "dose effect" of PCV—for each decile increase in coverage, there was a decline in pneumococcal disease, and, with the exception of parental smoking, the prevalence of measured confounders changed very little over time. Our definition of undervaccinated included children who received 1 dose of vaccine under 12 months of age who may benefit from some direct protection; however, majority of undervaccinated person-time was in children who had no doses (83%), and sensitivity analyses among completely unvaccinated children yielded similar results [38,39]. We consider the risk of misclassification of vaccination status to be low, since vaccination status was determined using an immunisation registry, which has previously been demonstrated to underestimate coverage by less than 5% [40]. Lastly, an important limitation of this study is the potential for loss to follow-up in children who have moved interstate (missing outcome data) or overseas (missing vaccination and outcome data). From 2001 to 2012, the annual interstate out-migration of children less than 5 years of age ranged from 1.6% to 2.2% in New South Wales and 1.4% to 2.1% in Western Australia [41]. Data available from 2004 to 2012 indicate that overseas out-migration was less common, ranging from 0.9% to 1.2% [42]. Since our analyses are for children up to 5 years of age, we estimate that the cumulative unobserved loss to follow-up time to be less than 15% —in line with recommendations for cohort [43]. Furthermore, we do not expect differential loss to follow-up between vaccinated and undervaccinated children or as vaccine coverage rises over time, since migration patterns have remained constant.

Our results support continuing efforts to increase vaccine coverage to maximise indirect protection from PCV in Australia. In our study, PCV7-type IPD continue to decline as coverage increases among children under 5 years, suggesting that catch-up vaccination up to 5 years of age may be important to maximise indirect effects—in line with previous research [44]. Indirect protection is important because it comprises a substantial component of overall impact, improves cost-effectiveness of the vaccine, and protects infants too young to be vaccinated and adults. While our study does not include indirect effects on adult disease, prior research suggests that in low-transmission settings, indirect effects in children are effectively translated to indirect effects in adults, since children are the primary drivers of pneumococcal transmission [45,46].

Understanding the coverage required to achieve indirect protection is of considerable global interest in the potential for reduced dose (1 + 1) schedules [6] as the success of this program relies on the continued generation of indirect protection for infants who have not received sufficient doses to mount individual-level protection [6]. Our findings suggest that lower rates of vaccine coverage may still be able to confer considerable indirect effects, and, therefore, a 1 + 1 schedule may be suitable despite low PCV coverage. Additionally, we found evidence that these trends are similar across a range of subgroups. However, our findings cannot be directly extrapolated to other settings since degree of indirect protection is highly dependent on local factors influencing pneumococcal transmission [28]. Other studies have demonstrated lower than expected direct and indirect effects following PCV13 introduction in Australia, which used a 3 + 0 schedule, compared to countries using a booster dose, such as the UK (2 + 1 schedule) and the United States (3 + 1 schedule) [17,47]. As a result, Australia implemented a switch from the 3 + 0 to 2 + 1 schedule in 2018. Further research beyond the time frames available in this analysis is needed to document the longer-term impacts of PCV13 introduction and the schedule change on indirect effects in Australia. Further research,

using consistent and robust methodology, is also required to understand the relationship between PCV coverage and indirect protection in a range of settings, particularly in high-transmission settings. This will assist with understanding the role of PCV coverage as an alternative metric to determine if indirect effects are present in settings with insufficient disease surveillance.

## Supporting information

**S1 RECORD Checklist. The RECORD statement—checklist of items, extended from the STROBE statement, which should be reported in observational studies using routinely collected health data.**
(DOCX)

**S1 Text. Protocol.**
(DOCX)

**S1 File. Supplementary methods and additional analyses.** Fig A. Flow chart of study cohort, including data sources. Fig B. Directed acyclic graph. IPD, invasive pneumococcal disease; LBW, low birth weight; No. siblings, number of siblings; PCV status, pneumococcal conjugate vaccination status; SLA PCV coverage, pneumococcal conjugate vaccine coverage within the statistical local area. Table A. Description of study covariates. Table B. 10th Revision of the International Classification of Diseases Australian Modification (ICD-10-AM) codes for classification of pneumonia hospitalisation. Table D. Rate of invasive pneumococcal disease (IPD) among all children and undervaccinated children by vaccination period age group and Indigenous status, New South Wales and Western Australia, 2001–2013. Table E. Annual pneumococcal conjugate vaccine coverage (PCV) and rate of 7-valent PCV-type invasive pneumococcal disease (IPD) among all children and undervaccinated children and among children under 5 years of age, New South Wales and Western Australia, 2005–2012. Table F. Annual 13-valent pneumococcal conjugate vaccine coverage (PCV13) and rate of PCV13, non PCV7-type invasive pneumococcal disease (IPD) among all children and undervaccinated children and among children under 5 years of age, New South Wales and Western Australia, 2005–2012. Table G. Annual incidence of pneumonia hospitalisation among all children and undervaccinated children by vaccination period, age group, and Indigenous status, New South Wales and Western Australia, 2001–2013. Table H. Crude rates of invasive pneumococcal disease (IPD) among undervaccinated children by level of pneumococcal conjugate vaccine (PCV) coverage within their statistical local area of residence, New South Wales and Western Australia, 2001–2012. Table I. Crude and adjusted* incidence rate ratios of 7-valent pneumococcal conjugate vaccine-type (PCV7-type) invasive pneumococcal disease (IPD) in different subgroups, by percent increase pneumococcal conjugate vaccine (PCV) coverage† among children under 5 years of age, New South Wales and Western Australia, 2009–2012. Table J. Crude and adjusted* incidence rate ratios of 7-valent pneumococcal conjugate vaccine-type (PCV7-type) invasive pneumococcal disease (IPD) among unvaccinated children under 5 years, by percent increase in pneumococcal conjugate vaccine (PCV) coverage† among children under 5 years of age, New South Wales and Western Australia, 2009–2012. Table K. Preventable fraction of PCV7-type IPD in unvaccinated children under 5 years at deciles of pneumococcal conjugate vaccine (PCV) coverage, New South Wales and Western Australia, 2001–2012. Table L. Rate of all-cause pneumonia hospitalisation among undervaccinated children by level of pneumococcal conjugate vaccine (PCV) coverage within their statistical local area of residence, New South Wales and Western Australia, 2001–2012 13. Table M. Crude and adjusted* incidence rate ratios of all-cause pneumonia in different subgroups, by percent

increase in pneumococcal conjugate vaccine (PCV) coverage† among children under 5 years of age, New South Wales and Western Australia, 2001–2012.
(DOCX)

## Acknowledgments

We thank the staff at the Population Health Research Network (PHRN), participating PHRN data linkage and infrastructure nodes (the Western Australian Data Linkage Branch, the New South Wales Centre for Health Record Linkage, and the Australian Institute for Health and Welfare), the Western Australia and Commonwealth Departments of Health and New South Wales Ministry of Health who provided advice and the data, and the Aboriginal and Torres Strait Islander community and members of the Aboriginal Immunisation Reference Group for their contribution to this research project.

## Author Contributions

**Conceptualization:** Jocelyn Chan, Fiona M. Russell.

**Data curation:** Parveen Fathima.

**Formal analysis:** Jocelyn Chan, Cattram D. Nguyen.

**Funding acquisition:** Heather F. Gidding, Hannah C. Moore.

**Methodology:** Jocelyn Chan, Heather F. Gidding, Peter B. McIntyre, Ross Andrews.

**Project administration:** Heather F. Gidding.

**Supervision:** Heather F. Gidding, Christopher C. Blyth, Kim Mulholland, Cattram D. Nguyen, Ross Andrews, Fiona M. Russell.

**Writing – original draft:** Jocelyn Chan.

**Writing – review & editing:** Heather F. Gidding, Christopher C. Blyth, Parveen Fathima, Sanjay Jayasinghe, Peter B. McIntyre, Hannah C. Moore, Kim Mulholland, Cattram D. Nguyen, Ross Andrews, Fiona M. Russell.

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
