## [Editor Report · Decision Letter 0]

10 Dec 2020

Dear Dr Chan, 

Thank you for submitting your manuscript entitled "Substantial indirect protection against invasive pneumococcal disease and pneumonia hospitalisations in Australian children at low levels of pneumococcal conjugate vaccine coverage: an observational study" for consideration by PLOS Medicine.

Your manuscript has now been evaluated by the PLOS Medicine editorial staff and I am writing to let you know that we would like to send your submission out for external peer review.

Kind regards,

Caitlin Moyer, Ph.D.,

Associate Editor

PLOS Medicine

---

## [Decision Letter · Decision Letter 1]

26 Apr 2021

Dear Dr. Chan,

Thank you very much for submitting your manuscript "Substantial indirect protection against invasive pneumococcal disease and pneumonia hospitalisations in Australian children at low levels of pneumococcal conjugate vaccine coverage: an observational study" (PMEDICINE-D-20-05700R1) for consideration at PLOS Medicine. 

Your paper was evaluated by a senior editor and discussed among all the editors here. It was also discussed with an academic editor with relevant expertise, and sent to three independent reviewers, including a statistical reviewer. The reviews are appended at the bottom of this email and any accompanying reviewer attachments can be seen via the link below:

[LINK]

In light of these reviews, I am afraid that we will not be able to accept the manuscript for publication in the journal in its current form, but we would like to consider a revised version that addresses the reviewers' and editors' comments. Obviously we cannot make any decision about publication until we have seen the revised manuscript and your response, and we plan to seek re-review by one or more of the reviewers. 

We expect to receive your revised manuscript by May 17 2021 11:59PM. Please email us (plosmedicine@plos.org) if you have any questions or concerns.

We look forward to receiving your revised manuscript. 

Sincerely,

Caitlin Moyer, Ph.D.

Associate Editor 

PLOS Medicine

plosmedicine.org

1.Title: Please revise your title according to PLOS Medicine's style. Your title must be nondeclarative and not a question. It should begin with main concept if possible. "Effect of" should be used only if causality can be inferred, i.e., for an RCT. Please place the study design ("A randomized controlled trial," "A retrospective study," "A modelling study," etc.) in the subtitle (ie, after a colon).

2. Financial disclosure: Please include the statement “The funders had no role in study design, data collection and analysis, decision to publish, or preparation of the manuscript” unless this does not apply.

3. Abstract: Please structure your abstract using the PLOS Medicine headings (Background, Methods and Findings, Conclusions).

4. Abstract: Methods and Findings: Please indicate in the abstract which two states of Australia are included.

5. Abstract: Methods and Findings: Please include the important variables that are adjusted for in the analyses.

6. Abstract: Methods and Findings: Please quantify the main results (with 95% CIs and p values).

7. Abstract: Methods and Findings: In the last sentence of the Abstract Methods and Findings section, please describe the main limitation(s) of the study's methodology.

8. Abstract: Conclusions: Please address the study implications without overreaching what can be concluded from the data; the phrase "In this study, we observed ..." may be useful.

9. Author Summary: At this stage, we ask that you include a short, non-technical Author Summary of your research to make findings accessible to a wide audience that includes both scientists and non-scientists. The Author Summary should immediately follow the Abstract in your revised manuscript. This text is subject to editorial change and should be distinct from the scientific abstract. Please see our author guidelines for more information: https://journals.plos.org/plosmedicine/s/revising-your-manuscript#loc-author-summary

10. Main text: Please use square brackets for in-text citations, like this [1]. For citations of multiple references, please do not include spaces within brackets.

11. Main text: Please include line numbers with the revised version.

12. Introduction: Paragraph 2: Please remove the trademark symbol from Prevnar, PNEUMOSIL, and Synflorix.

13. Methods: Please ensure that the study is reported according to the STROBE guideline, and include the completed STROBE checklist as Supporting Information. Please add the following statement, or similar, to the Methods: "This study is reported as per the Strengthening the Reporting of Observational Studies in Epidemiology (STROBE) guideline (S1 Checklist)."

14. Methods: Did your study have a prospective protocol or analysis plan? Please state this (either way) early in the Methods section.

15. Method: “Infant pneumococcal vaccination in Australia” Please consider whether this section may be better incorporated into the Introduction.

16. Results: Please present a table of summary demographic characteristics for the cohort.

17. Results: Please provide both 95% CIs and p values wherever relevant alongside results presented in the text; for example, when describing the associations between PCV coverage and PCV7 and PCV13-type IPV/pneumonia incidence.

18. Discussion: Please present and organize the Discussion as follows: a short, clear summary of the article's findings; what the study adds to existing research and where and why the results may differ from previous research; strengths and limitations of the study; implications and next steps for research, clinical practice, and/or public policy; one-paragraph conclusion.

19. References: Please use the "Vancouver" style for reference formatting, and see our website for other reference guidelines https://journals.plos.org/plosmedicine/s/submission-guidelines#loc-references

20. Figures 4, 5, and 7: please indicate in the legends that the blue shaded area represents the 95% confidence interval.

Comments from the reviewers:

Reviewer #1: The article by Chan et al describes an assessment of indirect protection derived from vaccination of children with pneumococcal conjugate vaccines in two States in Australia. Using a large research dataset linking birth certificates, vaccination records, laboratory surveillance results, and hospitalization data, investigators designed a retrospective cohort study to examine the incidence of invasive pneumococcal disease and pneumonia hospitalizations among children younger than 5 years. Incidence estimates were compared across different vaccination period, based on vaccine formulations and vaccination schedules. Incidence estimates were generated separately for all children and for under-vaccinated children, and assessments among under-vaccinated children were used to investigate the incidence of study outcomes by area vaccinate coverage estimates for an assessment of indirect protection. The investigators report a substantial indirect protection among under-vaccinated children derived from vaccination of young children with pneumococcal conjugate vaccines, with declining disease incidence with increasing vaccination coverage in the children's resident area. Material reductions were reported even with vaccination coverage of ~50%, and substantial benefits were calculated based on the estimated benefit of vaccination. 

Understanding the critical coverage of vaccination needed to attain population level indirect protection is of great public health interest. Yet, this is rarely examined using actual individual level data. These assessments are particularly important for expensive vaccines such as pneumococcal conjugate vaccines. So, this is a very important and relevant study. Linking multiple available databases, investigators have designed a powerful research data platform to address the questions of interest. The article is very well written, and the assessments are clearly described. This is a very elegant study with very important public health implications. However, I have a few comments/questions:

It is unclear how losses to follow-up were identified in the retrospective cohort study. This can be problematic if children leaving the study area were less likely to complete vaccination and if their outcomes were also not recorded in the source databases. Per the description in the article, based on birth certificates - everyone identified was assumed to remain under observation, but I am not sure this was actually demonstrated or discussed. It would be useful to conduct an exploration of an unrelated outcome (i.e. unlikely to be affected by vaccination) to ensure differential losses to follow-up were not a problem in the study. Alternative sources of follow-up information from other regions of the country could be useful as well. This potential issue should be addressed and discussed.

It would be very valuable to provide some additional description and references to substantiate the validity of the vaccination registry data - if vaccinated children somehow had their vaccination records affected they may appear as under-vaccinated when they were indeed fully vaccinated. This important misclassification possibility should be discussed in more detail as well. 

Previous related studies (see work by Loughlin et al [ref 30] and by Hammit et al. Lancet Global Health 2014) have estimated that approximately 70% coverage would be needed to attain indirect protection against carriage of vaccine types. The apparent discrepancy is intriguing as carriage is the recognized precursor for pneumococcal diseases. Reconciling those previous findings with the current estimates would be a valuable addition to the discussion section. 

The relatively limited impact of vaccination coverage in rural and indigenous populations is very intriguing and needs more discussion. The lack of significant associations among younger infants is intriguing as well. In part, these observations were attributed to limited social mixing in rural population relative to urban populations. However, ref [21] suggest there are more intense contacts in rural indigenous settings (e.g. facilitated by crowding and other factors) which facilitates spread of infections. The discussion needs to clarify and reconcile these elements. Also, the sample sizes were relatively small for some groups, but the rural subgroup was not that small. So other factors, separate from limited statistical precision, may potentially play a role in the current findings. 

Some covariates appear to be listed in the DAG figure, but it is not clear if those were accounted for in the regression analyses (e.g. number of children/siblings in the house, flu vaccination, etc.). Please reconcile the narratives and clarify/revise accordingly.

Reviewer #2: This retrospective population-based cohort study, comprising a subset of children under five years of age from a dataset of all children born in New South Wales (NSW) and Western Australia (WA) between 2001 and 2012, investigates the association between population PCV coverage and indirect protection against invasive pneumococcal disease and pneumonia hospitalisations among under-vaccinated Australian children.

Comments:

"To visualise any geographic heterogeneity in PCV coverage, we mapped coverage within Statistical Local Areas among children under five years of age at three time points - December 2005, December 2009, December 2012"

Can the authors please add an explanation here to clarify why these time points were chosen to map coverage? 

The use of Poisson regression models is technically appropriate, and the inclusion of various potential confounding variables is robust. Furthermore, suitable subgroup and sensitivity analyses have been included in the study.

Typo in: " Person-time for each child started at birth and was censored at the earliest of the following: death, when the child reached five years of age or at the end of the of the study period."

Figure 4 typo: A and B (currently both labelled as B)

Overall, this is a very well written report within which the main study limitations have been fairly acknowledged. It also comes with an extensive and thorough analysis contained within the supplementary material.

Reviewer #3: Substantial indirect protection against invasive pneumococcal disease and pneumonia

hospitalisations in Australian children at low levels of pneumococcal conjugate vaccine

coverage: an observational study

This study used IPD incidence and serotype data from state-based passive surveillance, Vaccine status from Australian Childhood Immunisation Registry and Hospitalisation data covering all inpatient separations to calculate the decline in VST disease, PCV7 or PCV13 compared to PCV7 and PCV13 uptake. Their strategy was to calculate the incidence of each disease outcome during the person-time for which children were under-vaccinated up to five years of age. They divided each child's under-vaccinated person-time into successive three-month intervals and linked these intervals to PCV coverage in their SLA of residence during the same time period. Thirty one percent of the childfen were among under-vaccinated or unvaccinated children, the majority of whom were unvaccinated. The conventional wisdom has been that reduction in colonization with VST Sp and indirect effect was observed beginning at ~ 50% vaccine penetration and increased with increasing vaccine uptake. The authors find lower level of community uptake, specifically in urban centers, has a larger impact than previously identified.

If correct, this would have important implications as high uptake, especially with dose after 12 months of age, may be more difficulty to achieve in certain setting.

Table 1 reports the declining incidence in immunized and unimmunized populations over intervals of time and the subsequent graph shows the dramatic increasing in vaccine uptake occurring over a few year. The decline per percent increase in vaccine uptake is then calculated. I found myself wanting to see more granular data; the incidence decline from 7.3 to 1.3 over a 11 year time line. I think showing the decline each year from 2001 through 2011 and the coverage each year for each group would improve clarity for the reader to view the uptake in the community and the decline in the vaccinated and under-vaccinated group. The rapid uptake as demonstrated in figure 3 does not allow the impact of the incremental uptake after introduction to be assessed as the data is presented for the 5 year period.

As vaccine uptake included a catch up through two years it is unclear if the age distribution in the comparison groups are comparable for each 3 month block assessed. Obviously there is an age specific risk for IPD and pneumonia and if the age distribution of each group differs then the incidence within those group may differ regardless of the vaccine impact. Is it possible the vaccinated group is likely biased toward children under 3, while the unvaccinated is biased toward children over 2 which would be a lower risk for IPD?

The graphs of vaccine uptake vs decline in IPD (figure 4 and 5) demonstrate substantially different relationships for PCV7 uptake and decline compared to PCV13 and decline. The authors explain this by difference in effectiveness against the PCV7 vs the PCV13 unique serotypes. As the vaccine is less effective against serotype 3, it would be of value to compare

With and without serotype 3 (for PCV13) as a sensitivity analysis to determine if the concept that 40% penetration is sufficient to produce substantial indirect benefit is limited to PCV7 or more broadly applicable to pneumococcal conjugate vaccines. And if limited to PCV7, why???

Figure 5 demonstrates that different populations have different dynamics. The indirect effect for those less than 4 months and Indigenous and rural individuals has a different relationship between vaccine uptake and indirect effect compared to urban environment. How does the disease epidemiology impact on this relationship? It appears that incidence in first 4 month is relatively low; that urban children have highest incidence; and Indigenous children are not as high as expected as I thought their rate of disease was highest in the population?? Are these relationships between vaccine uptake and disease decline similar for direct effect or only for indirect effect?

Other comments:

The UK introduction of 1+1 regimens was after nearly 10 years of PCv13 use and little residual VST circulation. If the authors want to discuss the potential value of 1+1, I would advise also adding potential limitations in this environment and need for high uptake of the booster dose to be implemented (as recommended by the UK).

The supplemental data was very valuable but did not provide specific information to address my questions above.

Overall, this is a very valuable analysis. The largest concern is how the age distributions of the immunized and unimmunized groups compare and whether some adjustment needs to be included as incidence is age sensitive in the first 5 years of life.

[LINK]

---

## [Decision Letter · Decision Letter 2]

7 Jul 2021

Dear Dr. Chan,

Thank you very much for re-submitting your manuscript "Pneumococcal conjugate vaccine coverage and indirect protection against invasive pneumococcal disease and pneumonia hospitalisations in Australia: a retrospective record linkage cohort study" (PMEDICINE-D-20-05700R2) for review by PLOS Medicine.

I have discussed the paper with my colleagues and the academic editor and it was also seen again by one of the original reviewers. I am pleased to say that provided the remaining editorial and production issues are dealt with we are planning to accept the paper for publication in the journal.

[LINK]

We look forward to receiving the revised manuscript by Jul 14 2021 11:59PM.   

Sincerely,

Caitlin Moyer, Ph.D.

Associate Editor 

PLOS Medicine

plosmedicine.org

Requests from Editors:

1. Title: We suggest revising to: “Levels of pneumococcal conjugate vaccine coverage and indirect protection against invasive pneumococcal disease and pneumonia hospitalisations in Australian children: An observational study”

2. Abstract: Please combine the Methods and Findings sections into one section, “Methods and findings”. Please move the sentence describing the main limitations to the end of this section.

3. Abstract: Methods and Findings: Line 16: Please clarify if “addressed” should be “residential address” or similar.

4. Abstract: Lines 22-24: We suggest revising to: “Fifty-percent coverage of PCV7 among children < five years of age was estimate to prevent up to 72.5% (95% CI 51.6-84.4) of PCV7-type IPD among under-vaccinated children, while 90% coverage was estimate to prevented 95.2% (95% CI 89.4-97.8).” or similar.

5. Abstract: Conclusions: Please expand on this sentence, for example, please clarify “...high vaccine coverage is required” by explaining what high coverage is meant to prevent. If space permits, we would suggest an additional sentence, perhaps discussing the implications of your findings for policy considering vaccination schedule, for example.

6. Author summary: Why was the study done? Please revise the third bullet point to: “In this study, we examined associations between PCV coverage and indirect effects within diverse populations within Australia.’

7. Methods: Page 5: Line 21-22: Please revise to “...defined as isolation of S. pneumoniae by culture or detection of nucleic acid from a normally sterile sample…” or similar, as a word seems to be missing.

8. Methods: Thank you for including the RECORD checklist. Please also add the following statement to the Methods: "This study is reported as per the REporting of studies Conducted using Observational Routinely-collected Data (RECORD) guideline (S1 Checklist)."

9. Results: Page 12: LIne 17-19: The term "trend" is used here to refer to a non-statistically significant P value. The term trend should be used only when the test for trend has been conducted. Please revise accordingly.

10. Results: For adjusted analyses of PCV coverage and incidence, please also confirm the unadjusted results are presented in the manuscript.

11. Discussion: Page 16: First paragraph: We suggest tempering the language in the paragraph slightly, for example: “Our novel analyses highlight the relatively low levels of coverage required to generate indirect protection in children against IPD and pneumonia hospitalisations in Australia.” could be revised to reflect that findings from these analyses support that a low level of coverage may be required for indirect protection, as compared to what has been previously thought. Also, “indirect effects” is used frequently here and throughout the manuscript. Please consider revising throughout to use an alternative term, or qualifying as “estimated indirect effects” to minimize implications of direct causal effects.

12. Discussion: Page 17: Line 25: Please include the in-text reference citations within square brackets. 

13. Discussion: Page 16: Line 13: Please place the reference bracket before the sentence punctuation.

14. Discussion: Page 18: Line 32: Please change to “...able to confer…” if that is intended.

15. Page 19: Please remove the “Funding” and “Contributors” sections from the main text, as these will be included with the metadata associated with the manuscript submission.

16. References: Please double check all references for the use of the "Vancouver" style for formatting (in particular, please check Journal title abbreviations, for example, in reference 1, Lancet Global Health should be Lancet Glob Health, and see our website for other reference guidelines: https://journals.plos.org/plosmedicine/s/submission-guidelines#loc-references

17. S1 Protocol: In the list of covariates the term “gender” is used, while “sex” is used in the main text (Table 1). The terms gender and sex are not interchangeable (as discussed in http://www.who.int/gender/whatisgender/en/ ); please use the appropriate term.

18. Table 1: For the category “sex” please clarify whether percentages are of the total number. For smoking, please provide the responses for “no” if informative.

19. Figure 2 and 3: Please provide a descriptive label for the color indicator key used.

20. Figure 2 and 3 (map images): Please confirm that the appropriate usage rights apply to the use of this map, and please replace with alternative images if necessary. Please see our guidelines for map images: https://journals.plos.org/plosmedicine/s/figures#loc-maps

21. SI file: Figure 2: In the legend, please include definitions for each abbreviation used in the figure.

22. Supporting information file: Table S10 and S11: Please include these two tables within the Results section (main text) of the paper, if possible, rather than in the supporting information files. 

Comments from Reviewers:

Reviewer #2: The authors have satisfactorily responded to each comment in turn.

[LINK]

---

## [Editor Report · Decision Letter 3]

13 Jul 2021

Dear Dr Chan, 

On behalf of my colleagues and the Academic Editor, Mirjam Kretzschmar, I am pleased to inform you that we have agreed to publish your manuscript "Levels of pneumococcal conjugate vaccine coverage and indirect protection against invasive pneumococcal disease and pneumonia hospitalisations in Australia: an observational study" (PMEDICINE-D-20-05700R3) in PLOS Medicine.

In addition, we request that you please capitalize the first word of the subtitle after the colon ("An observational study").

PRESS

Sincerely, 

Caitlin Moyer, Ph.D. 

Associate Editor 

PLOS Medicine